# A Note on Matricial Ways to Compute Burt's Structural Holes

Alessio Muscillo 

Department of Economics and Statistics, Università di Siena, 53100 Siena, Italy; alessio.muscillo2@unisi.it

**Abstract:** In this note, I derive simple formulas based on the adjacency matrix of a network to compute measures associated with Ronald S. Burt's structural holes (effective size, redundancy, local constraint, and constraint), together with the measure called *improved structural holes* introduced in 2017. This can help to see these measures within a unified computation framework because they can all be expressed in matricial form. These formulas can also be used to define naïve algorithms based on matrix operations for their computation. Such naïve algorithms can be used for small- and medium-sized networks, where exploiting the sparsity of the matrices and efficient triangle listing techniques are not necessary.

**Keywords:** network measures; structural holes; effective size; redundancy; constraint

## 1. Introduction

The notion of Burt's structural holes, used when analyzing social networks, is pervasive and fascinating [1]. Intuitively, it refers to the absence of connections between groups and has been linked to the fact that filling these voids and bridging these gaps can be a source of opportunities and be very beneficial for individuals able to do so. On the one hand, the versatility of this concept has stimulated the definition of several measures, each capturing a different aspect [2], and each measure has been used in different frameworks when analyzing real-world networks and social capital theories [3]. On the other hand, this has also generated confusion when it comes to which exact measure one has to compute and how to compute it [4,5]. Moreover, computing these measures directly by applying the definition formulas can be very slow and computationally intensive, because it would require looping over each node's neighbors (and its neighbors' neighbors).

As clearly described by [6], the advantages of structural holes lie in the information benefits and in the control benefits. The former refer to the possibility of acquiring information from different communities where there may be different opinions, ideas, and pieces of information. The latter, instead, refer to the better negotiating position that a *structural hole spanner* has because she has access to unique pieces of information with respect to other agents who are instead closed and confined within the groups that are being bridged. In several applications, it has been shown that structural holes are important for channeling information flows, and in this way they can bridge different groups that would otherwise not communicate with each other, thus acting as brokers. However, by blocking and controlling such information flow, they play a crucial role in shaping opinion dynamics, since they can also maintain close communities unaware of other groups' opinions and beliefs, possibly exacerbating echo-chamber effects [7–9]. In a world dominated by online social networks, studying these phenomena can have important applications when opinions about public and health issues are formed and spread, as recently demonstrated by the different and conflicting ideas about the COVID-19 vaccination campaigns that circulated in restricted groups of the population [10,11].

In this note, I consider the main measures associated with Burt's structural holes, namely effective size, redundancy, local constraint, and constraint, and derive simple formulas for them based on the adjacency matrix of the network. This can help to have a unified framework where all measures are not only computed starting from the adjacency matrix, but also with a matrix representation. It can also help to interpret and compare these measures and produces intuitive and naïve algorithms, based on matrix multiplications, that work fairly well on medium-sized networks (see Section 7 for more details on this). While this approach is clean and simple, it has clear limitations when the network being analyzed becomes very large. In such a case, one should avoid storing the matrices explicitly and rely on distributed algorithms and more advanced techniques for triangle listing in sparse graphs. It should be noted here that one of the limiting factors of this paper's simple approach is that the matrix $A^2$ is denser than the adjacency matrix $A$. A comparison in a controlled environment should show that more efficient algorithms based on triangle listing with vertex orderings and neighborhood markers [12] scale better with the number of nodes than the naïve algorithms proposed here.

Thus, the contribution of this paper to the extant literature consists, first, in making clear that all these measures can be easily and directly computed starting from the network's adjacency matrix and, secondly, that the formulas obtained can be used as naïve algorithms that can work when applied on networks of medium dimension without the use of advanced techniques.

The paper is structured as follows. In Section 2, we clarify the notation that will be used in the rest of the article, especially for what concerns matrix operations and element-wise operations. In Section 3, we start from the definition of *effective size* (and *redundancy*) given by [4] and show how it can be written with vector- and matrix-based operations (i.e., Equation (4)). With the same line of reasoning, in Sections 4 and 5, we start from the definition of *local constraint* and of *constraint* given by [5] and show how they too can be written with vector- and matrix-based operations, respectively in Equations (16) and (17). Then, in Section 6, we adopt the same approach and show how the measure developed by [13] called *improved structural holes* can be written with matrix–vector operations, as in Equation (18). Lastly, Section 7 concludes the paper.

## 2. Notation

In what follows, matrices are denoted by capital letters (e.g., $A$ and $P$) and their elements denoted by the corresponding letter with subscripts (e.g., $a_{ij}$ and $p_{ij}$). Generic nodes of a network (i.e., a graph) will be indicated by $i$, $j$, or $k$. Consequently, an adjacency matrix will be indicated by $A = (a_{ij})_{i,j=1,\ldots,n}$, where $n$ is the number of nodes and the elements $a_{ij}$ can be 0 or 1 for binary networks or generic real numbers for weighted networks.

Vectors and their elements will be respectively denoted by bold letters (e.g., $\mathbf{x}$ and $\mathbf{y}$) and letters with a (single) subscript (e.g., $x_i$). The vector obtained by taking the diagonal elements of a square matrix $A$ is denoted by $Diag(A)$; analogously, the matrix that has $\mathbf{x}$ as its diagonal and 0s elsewhere is denoted by $Diag(\mathbf{x})$. The transposed of a vector or matrix is denoted by $\cdot^T$ (e.g., $\mathbf{x}^T$ and $A^T$). Hereafter, vectors are considered as columns, that is $(n \times 1)$-matrices and their transposed as row vectors $\mathbf{x}^T$. Accordingly, the (matrix) multiplication of a column vector $\mathbf{x}$ times a row vector $\mathbf{y}^T$ will yield a matrix (e.g., $\mathbf{x}\mathbf{y}^T \in \mathbb{R}^{n \times n}$), whereas $\mathbf{x}^T\mathbf{y}$ will yield a scalar.

The matrix multiplication between two matrices $A$ and $B$ will be denoted by juxtaposition, i.e., $AB$, whereas element-by-element operations such as element-wise multiplication or division will be denoted, respectively, by $\odot$ and $\oslash$. The $n$-dimensional unitary vectors in $\mathbb{R}^n$ containing all 0s but one 1 in the $i$-th position is denoted by $\mathbf{e}_i$, while the vector containing all 1s is denoted by $\mathbf{1}$. The identity matrix is denoted by $I$.

## 3. Effective Size and Redundancy for Undirected Binary Networks

The original definition of effective size and redundancy in Burt's works was complicated, but Borgatti [4] has shown that it can be simplified. Here, we consider an undirected and binary network with no self-loops. The intuitive idea (see Figure 1) is first to compute a node's *redundancy*, which is the mean number of connections from a neighbor to other neighbors. The *effective size* is then obtained by subtracting the redundancy to the node's degree.

**Description of redundancy and effective size**

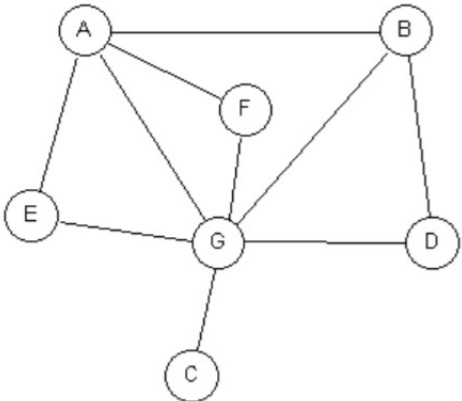

**Figure 1.** Adapted from Burt's and Borgatti's works. Let us compute the effective size for node *A*. Consider one of its neighbors, say *G*. They have 3 "common neighbors": *B*, *E*, and *F*. Divide this number by *A*'s degree: 3/4. Take another of *A*'s neighbors, say *E*. *A* and *E* have just 1 common neighbor, which is *G*. Therefore, in this case, dividing by *A*'s degree gives 1/4. To determine *A*'s *redundancy*, repeat this process for all 4 neighbors of *A* (respectively, *B*, *E*, *F*, and *G*) and sum up all the numbers obtained: $\frac{1}{4} + \frac{1}{4} + \frac{1}{4} + \frac{3}{4} = \frac{6}{4} = 1.5$. Lastly, *A*'s effective size is its degree minus its redundancy: $4 - 1.5 = 2.5$. In the example of this section, the matricial computation is also done for the remaining nodes.

Let $r_i$ be the redundancy of node $i$ and let $t_i$ be "the number of ties in the network (not including ties to ego)" [4] (Note: "ego" here is node $i$). The redundancy is then simply (since the network is assumed undirected, the links of $t_i$ have to be counted twice)

$$r_i = \frac{2\,t_i}{d_i}, \tag{1}$$

where $d_i$ is $i$'s degree. Notice that $r_i$ goes from 0 to $d_i - 1$. (This is also well related to the notion of *local clustering*, which can be thought of as a normalized version of redundancy ranging from 0 to 1. It can easily be shown that the relationship between the local clustering $q_i$ and redundancy $r_i$ of a node $i$ of degree $d_i$ is given by $q_i = \frac{r_i}{d_i - 1}$ [14]). The effective size $s_i$ of node $i$ is then defined as

$$s_i = d_i - r_i. \tag{2}$$

We now look at how to compute this in a matricial form. Let $A = (a_{ij})_{i,j}$ be the adjacency matrix of such an undirected and binary network, and let $\mathbf{d} = (d_i)_i$ be the vector of nodes' degrees. It should be noted that $A$ is a symmetric matrix only containing 0s and 1s. In such a case the vector of nodes' degree can be obtained in several ways, for example, as $\mathbf{d} = Diag(A^2)$ or $\mathbf{d} = A\mathbf{1}$. Notice also that, for a binary network, the elements of the square $A^2$ count the number of common neighbors. Indeed, for every two nodes $i$ and $j$, the $(i, j)$-th element of $A^2$ is

$$(A^2)_{ij} = \sum_{k=1}^{n} a_{ik}a_{kj} = |\{k : k \in N(i) \text{ and } k \in N(j)\}| \tag{3}$$

$$= \text{ number of common neighbors of } i \text{ and } j,$$

since $a_{ik}$ is different from 0 if and only if $i$ and $k$ are linked; analogously, $a_{kj}$ is different from 0 if and only if $k$ and $j$ are linked. Obviously, we only want to count the common neighbors for pairs of nodes that are actually linked in the network. To do so, it suffices to multiply $A^2$ element by element for $A$ itself. Lastly, we want to sum all these numbers and divide them by the corresponding degree.

Summing up, a matricial way to compute the vector of nodes' effective size, $\mathbf{s} = (s_i)_i$, is by computing the following vector:

$$\mathbf{s} = \mathbf{d} - (A^2 \odot A)\mathbf{1} \oslash \mathbf{d}, \tag{4}$$

where $A^2$ is $A$ squared with the standard matrix multiplication. The $i$-th component of such a vector, $s_i$, is node $i$'s effective size. By definition, the redundancy is just the last term, that is $\mathbf{r} = (A^2 \odot A)\mathbf{1} \oslash \mathbf{d}$, where $\mathbf{r} = (r_i)_i$.

*An Example*

Consider the network in Figure 1. The adjacency matrix and the degree vector are, respectively,

$$A = \begin{pmatrix} . & 1 & . & . & 1 & 1 & 1 \\ 1 & . & . & 1 & . & . & 1 \\ . & . & . & . & . & . & 1 \\ . & 1 & . & . & . & . & 1 \\ 1 & . & . & . & . & . & 1 \\ 1 & . & . & . & . & . & 1 \\ 1 & 1 & 1 & 1 & 1 & 1 & . \end{pmatrix}, \quad \mathbf{d} = \begin{pmatrix} 4 \\ 3 \\ 1 \\ 2 \\ 2 \\ 2 \\ 6 \end{pmatrix}.$$

For simplicity, in $A$, the 0s are not indicated. Notice that self loops are not allowed.

Now, since

$$A^2 = \begin{pmatrix} 4 & 1 & 1 & 2 & 1 & 1 & 3 \\ 1 & 3 & 1 & 1 & 2 & 2 & 2 \\ 1 & 1 & 1 & 1 & 1 & 1 & 0 \\ 2 & 1 & 1 & 2 & 1 & 1 & 1 \\ 1 & 2 & 1 & 1 & 2 & 2 & 1 \\ 1 & 2 & 1 & 1 & 2 & 2 & 1 \\ 3 & 2 & 0 & 1 & 1 & 1 & 6 \end{pmatrix},$$

Equation (4) yields the effective size for each node:

$$\text{nodes} \left\{ \begin{array}{c} A \\ B \\ C \\ D \\ E \\ F \\ G \end{array} \right\} \rightarrow \begin{pmatrix} 2.5 \\ 1.667 \\ 1 \\ 1 \\ 1 \\ 1 \\ 4.667 \end{pmatrix}.$$

## 4. Local Constraint (a.k.a. Dyadic Constraint)

In this section, we adopt the same approach used in the previous section to obtain a formula based on the network's adjacency matrix for the so-called local constraint.

Let $A = (a_{ij})_{i,j}$ be the adjacency matrix of a network (not necessarily binary or unweighted). That is, $A$ is not necessarily symmetric and may contain elements different from 0 and 1. The only assumption here is that no self-loop is allowed, that is, $a_{ii} = 0$ for all nodes $i$. Following Everett and Borgatti [5], the *local constraint* on $i$ with respect to $j$, denoted $\ell_{ij}$, is defined by

$$\ell_{ij} = \left( p_{ij} + \sum_{k \in N(i) \setminus \{j\}} p_{ik} p_{kj} \right)^2, \tag{5}$$

where $N(i)$ is the set of neighbors of $i$, and $p_{ij}$ is the *normalized mutual weight* of the edges joining $i$ and $j$, that is,

$$p_{ij} = \frac{a_{ij} + a_{ji}}{\sum_k (a_{ik} + a_{ki})}. \tag{6}$$

This is also known as the *dyadic constraint*. Notice that assuming the absence of self-loops, then every $p_{ii} = 0$ because $a_{ii} = 0$. This implies that the second term in Definition (5) can be written as

$$\sum_{k \in N(i) \setminus \{j\}} p_{ik} p_{kj} = \sum_{k \in N(i)} p_{ik} p_{kj} - p_{ij} \underbrace{p_{jj}}_{=0} = \sum_{k \in N(i)} p_{ik} p_{kj}, \tag{7}$$

and, hence, $\ell_{ij}$ becomes

$$\ell_{ij} = \left( p_{ij} + \sum_{k \in N(i)} p_{ik} p_{kj} \right)^2. \tag{8}$$

Now, let us focus on $p_{ij}$, writing Equation (6) in matricial terms:

$$p_{ij} = (A + A^T)_{ij} \cdot \frac{1}{((A^T + A)\mathbf{e}_i)^T \mathbf{1}}, \tag{9}$$

and let us define vector $\mathbf{x} = (x_i)_i$, where

$$x_i = (A + A^T)\mathbf{e}_i^T \mathbf{1} = \sum_k (a_{ik} + a_{ki}), \tag{10}$$

Notice that the denominator here is the multiplication of a row vector times a column vector, which is a number.

Thus,

$$\mathbf{x} = (A + A^T)^T \mathbf{1} = (A^T + A)\mathbf{1} \tag{11}$$

and we can consider the vector containing all inverted elements:

$$\mathbf{y} = \mathbf{1} \oslash \mathbf{x} = \begin{pmatrix} 1/x_1 \\ \vdots \\ 1/x_n \end{pmatrix}. \tag{12}$$

Notice that $A^T + A$ is always symmetric, even if $A$ is not.

We then define a matrix that only requires that the diagonal is equal to $\mathbf{y}$, that is, $Diag(\mathbf{y})$. Now, we can finally compute $P = (p_{ij})_{i,j}$ as follows:

$$P = Diag(\mathbf{y}) (A + A^T). \tag{13}$$

By pre-multiplying a diagonal matrix, we are simply multiplying every row $i$ of $(A + A^T)$ for the corresponding element $y_i$ of the diagonal.

We will now focus on $\ell_{ij}$. Consider again the second term of the definition's formula as written in Equation (8)

$$\sum_{k \in N(i)} p_{ik} p_{kj} = \sum_k a_{ik} p_{ik} p_{kj}, \tag{14}$$

where the summation on the right-hand side is over all nodes $k$ (not just limited to $i$'s neighbors). Note that, if the network is weighted, then here one has to first compute the

binary version $A$ of the weighted adjacency matrix $W$, where $a_{ij} = 1$ if and only if $w_{ij} \neq 0$; otherwise, $a_{ij} = 0$. One can then simply apply the formula written in the text.

Written in matricial form, this summation in Equation (14) can simply be expressed as

$$(A \odot P)P, \tag{15}$$

where $\odot$ is the element-wise matricial multiplication and the second is a matrix multiplication. The modification regarding the dyadic constraint mentioned above consists in taking $P(P \odot A)$ here.

To conclude, we can write the matrix $L = (\ell_{ij})_{i,j}$ containing all links' local constraints as follows:

$$L = \big[ P + (A \odot P)P \big] \odot \big[ P + (A \odot P)P \big]. \tag{16}$$

Summing up, the algorithm to compute the local constraints with this formula takes the adjacency matrix $A$ as input and proceeds with the following steps:

1. $\mathbf{x} = (A + A^T)\mathbf{1}$;
2. $\mathbf{y} = \mathbf{1} \oslash \mathbf{x}$;
3. $P = Diag(\mathbf{y})\,(A + A^T)$;
4. $L = \big[ P + (A \odot P)P \big] \odot \big[ P + (A \odot P)P \big]$.

Clearly, the output of such algorithm is the matrix $L = (l_{ij})_{i,j}$ where $l_{ij}$ is the local constraint of link $ij$.

## 5. Constraint

In line with what was done in the previous sections, here we obtain a matrix-based formula to compute the network's constraint, starting from the local constraint computed in Section 4.

Let $L = (\ell_{ij})_{i,j}$ be the local constraint matrix computed in Equation (16). According to Everett and Borgatti [5], the *constraint* for node $i$ (also known as *first-type constraint* in the terminology used by [3]) is

$$c_i = \sum_{j \in N(i)} \ell_{ij}.$$

Notice that, in our notation, $N(i)$ does not include $i$ itself. To be even more clear, one could then write $c_i = \sum_{j \in N(i) \setminus \{i\}} \ell_{ij}$.

One can re-write this as follows:

$$c_i = \sum_j \ell_{ij} a_{ij},$$

where $(a_{ij})_{i,j} = A$ is the adjacency matrix. In case the network is weighted, then the matrix $A$ here is the binary version of the weighted adjacency matrix $W$, as observed in Footnote Section 4.

Therefore, the vector $\mathbf{c} = (c_i)_i$ containing the constraints of the network is obtained by summing the rows of the matrix $L \odot A$:

$$\mathbf{c} = \Big[ \mathbf{1}^T (L \odot A) \Big]^T. \tag{17}$$

Remember that, in our notation, vectors are always considered as columns.

Thus, one can use this matrix Formula (17) as an algorithm to compute the constraints of the network gathered in vector $\mathbf{c} = (c_i)_i$, where $c_i$ is node $i$'s constraint.

## 6. Improved Structural Holes

In this section, we adopt the same approach used so far and apply it to a variation of the constraint measure called *improved structural holes* (ISH) developed by [13]. This measure has been used to identify key nodes in networks and, with respect to measures of centrality such as betweenness and closeness, it has the advantage that it only uses local

information, while, in comparison with more local measures of centrality, such as degree centrality, it has the advantage that it is better able to capture the importance of a node. We now describe how to compute the ISH with a matrix-based formula.

Consistently with the notation used so far, let us follow [13] and consider their definitions.

- The *edge weight* between two connected nodes $i$ and $j$ is $w_{ij} := d_i + d_j$. In matricial form,

$$W = A \odot \left( \mathbf{d1}^T + \mathbf{1d}^T \right) = A \odot \left( A\mathbf{11}^T + \mathbf{11}^T A^T \right).$$

- The *node weight* is $w_i := \sum_{j \in N(i)} w_{ij}$, so that $\mathbf{w} = W\mathbf{1} \in \mathbb{R}^n$.
- The *relative importance* is $q_{ij} := \frac{w_{ij}}{w_i}$, with $\sum_j q_{ij} = 1$, and can be represented as

$$Q = W \oslash (W\mathbf{11}^T).$$

The *node importance* (also known as ISH) of $i$ is then defined as

$$s_i = \sum_{j=1}^{n} \left( q_{ij} + \sum_{k \in N(i) \setminus \{j\}} q_{ik}\, q_{kj} \right)^2, \quad \forall i = 1, \ldots, n.$$

In matricial form, the term within parenthesis $b_{ij} = q_{ij} + \sum_k a_{ij} q_{ik} q_{kj}$, where now $k$ is free to vary from 1 to $n$, and can be written as

$$B = Q + A \odot Q^2.$$

Thus, the vector of the ISH, $(ISH_1, \ldots, ISH_n) \in \mathbb{R}^n$, can be computed as

$$\mathbf{ISH} = (B \odot B)\mathbf{1}. \tag{18}$$

Concerning the computational complexity, as already observed by [13], the use of naïve matrix multiplications to compute the ISH would result in a complexity of the order $O(n^3)$; however, a deeper analysis can show that it can be computed with a time complexity of $O(n\langle d_i^2 \rangle)$, where $\langle d_i^2 \rangle$ is the average of the degrees squared.

## 7. Discussion and Conclusions

The notion of structural holes, proposed in Burt's seminal work [15], has been widely used to explain how, in networks where information flows through contacts, certain positions matter more than others. Individuals classified as structural holes spanners are able to exploit the lack of connections between separate parts of the network by bridging them and act as gatekeepers or brokers. These concepts have seen applications in the literature of management, sociology, and organization science, where the majority of the works conclude that where such network gaps are present, the role played by intermediaries gains importance precisely because they can control the flow of information and, hence, gain a competitive advantage from their position [16–20].

Structural holes theory is related to [21], which is on the strength of weak ties, because weak ties are capable of connecting parts of the network that would otherwise remain distant. It is also often seen as a competing theory with respect to [22], which is on the importance of network closure, redundancy, and network density [23]. However, these theories ultimately rely on the same underlying principle: ties alter the constraint ego [24].

Although the notion of structural holes is widespread and very often used, its actual measurement may be complicated for several reasons: First, a variety of measures have been proposed, and each one is tailored to a specific application, whose purpose is to capture a specific aspect of the structural hole notion. Second, as it often occurs for network measures, their actual computation with naïve techniques obtained from the definition formulas would require looping over every node and over every node's neighbor.

This paper thus contributes to the previous literature by making clear that all the measures commonly associated with the notion of structural holes can indeed be computed starting from the network's adjacency matrix, and we provide formulas that are based on this matrix and mainly use only matrix multiplication. As a consequence, we additionally show that these formulas can be used as naïve algorithms and directly applied to networks of up to a medium dimension to compute such structural holes measures. We also highlight the limitations of this approach that can in fact be used on moderately-sized networks (see Figure 2), but that should be outperformed by algorithms based on more advanced techniques that optimize memory and speed, although further research is still needed to explore this aspect.

**Example: a comparison with NetworkX's routines**

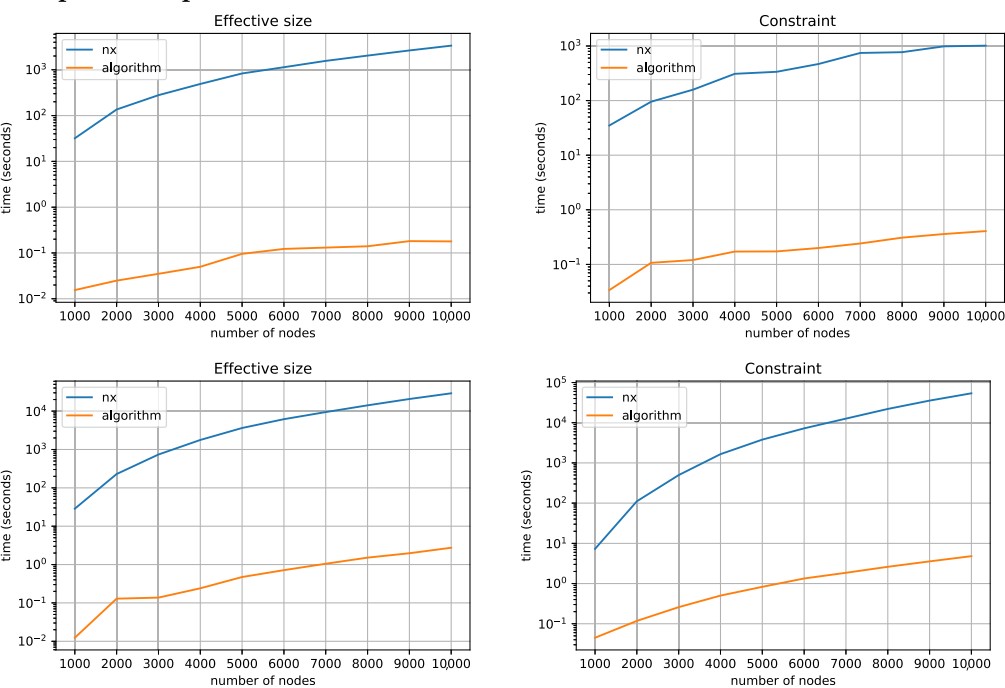

**Figure 2.** As an example, we perform a basic comparison of computational speed between this paper's algorithms (orange) and NetworkX's algorithms (blue) for effective size (left) and constraint (right), as the number of nodes increases from 1000 to 10,000. The networks are Barabasi-Albert graphs with parameter $m = 5$ (top row) and Erdos-Renyi random graphs with parameter $p = 0.01$ (bottom row). In particular, in orange, effective size and constraint are, respectively, computed using the formulas in Equations (4) and (17). All of the code to implement it is available here: https://github.com/alessiomuscillo/matricial_ways_for_Burt_structural_holes (accessed on 27 October 2022). However, a more precise comparison in a controlled environment should show that more efficient methods could scale better and be better suited for large networks. (All links accessed on 27 October 2022).

**Funding:** The author acknowledges funding from the 2020-2023 PRIN project 2017ELHNNJ "The economics of vaccination", financed by the Italian Ministry of Research and from the Open Access fund of the University of Siena.

**Institutional Review Board Statement:** Not applicable.

**Informed Consent Statement:** Not applicable.

**Data Availability Statement:** All of the code to replicate this study is available here: https://github.com/alessiomuscillo/matricial_ways_for_Burt_structural_holes (accessed on 27 October 2022).

**Acknowledgments:** The author thanks Tiziano Razzolini and Paolo Pin for their support and Martin Everett for his useful comments.

**Conflicts of Interest:** The author declares no conflict of interest.

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
