# Peer review of "A Note on Matricial Ways to Compute Burt’s Structural Holes"

_symmetry, doi:10.3390/sym15010211_

Round 1
Reviewer 1 Report
Burt's structural specified in the paper representing the author's work
Expressions and descriptions of holes, namely effective size, redundancy, local constraint and constraint are well-formed. However, the merits of this paper discussed in the summary are not yet fully described, and at the same time, the results described in Fig. 1 in Chapter 2 seem to be talking about the test results of Burt's structural holes, not the part to be described in Notation.
Furthermore, Chapters 4 and 5, which include the content discussed in Chapter 3, also do not explain the connection, so it is a very insufficient explanation as an academic paper. The question of whether the final chapter 6 has been improved has not been resolved.
Author Response
I report here what I have written in the PDF file uploaded with the reply to the Editor and the Reviewers.
At this link, you can find a version of the revised manuscript that is the same one that I have resubmitted, but where all the new parts are highlighted in a different color for readability.
ANSWER TO REVIEWER 1:
I thank the Referee for their comments. In the revised version, I expanded the "Introduction" section and added a last section "Discussion and conclusion" where, in particular, I describe more in detail the contributions of this paper in light of the previous literature.
Moreover, at the beginning of Sections 4, 5 and 6, I added introductory paragraphs to make each piece of the paper well connected with the rest.

Reviewer 2 Report
Please, see attached my comments.

Author Response
I report here what I have written in the PDF file uploaded with the reply to the Editor and the Reviewers.
At this link, you can find a version of the revised manuscript that is the same one that I have resubmitted, but where all the new parts are highlighted in a different color for readability.
ANSWER TO REVIEWER 2:
I thank the Referee for the precious suggestions and comments. In the current version, I added several pieces following the Referee's suggestions. In particular:
- as suggested by the Referee, in the "Introduction" section I added more detail to explain and clarify the paper's contribution to the previous literature. Moreover, I also added a last paragraph with the structure of the paper.
- Again, as suggested, at the end of the paper I also added a new section 6 "Discussion and conclusion".

Reviewer 3 Report
Thanks for the opportunity to read the paper. I enjoyed it.
I would like to request the author to discuss, in the Conclusions, some managerial aspects related to the findings, such as the gains people could enjoy from their connections under the perspective of the Structural Hole Theory.
Additionally, author could bring a historical perspective of the theory (ref.: Granovetter, 1973).
Author Response
I thank the Referee for their useful comments.
In the file attached the Referee can see my reply to their suggestions.
At this link, you may find a revised version of the manuscript where all the changes are highlighted in a different color for readability.

Round 2
Reviewer 1 Report
The results related to the graphs of the author's claims are discussed in the introductory part, showing very clear results. However, due to the sudden change in logic (each chapters and even meaning), it is very difficult for readers to understand.
In order to show the results related to the graph, the necessity and justification for why it has to come out like this is very lacking.
There is a jump in the overall logic, and the composition and format of the thesis (the method of giving references) is recommended to modify the method of engineering journal paper.
Author Response
I thank the Reviewer for their suggestions. I hope that they can see the uploaded file where I answer to the points raised by them.
Moreover, a revised version of the paper (where changed parts are in "blue" for readability) is available at this link.

Round 3
Reviewer 1 Report
The abstract part has not been updated yet and many parts are still ambiguous. Also, the abstract part does not show the important results that this paper talks about.
I am not sure that why the Introduction part presents the results of comparison and explains them. It is better to explain in the Discussion section or the Results section.
The constraints in Chapters 4 and 5 are vague and lack an explanation of how they affect the overall algorithm.
Evaluation is difficult because it does not follow the process of proof of the part discussed in Chapter 6, whether inductive or deductive.
Author Response
I thank the Reviewer for the effort and for the useful suggestions.
In the file attached, I provide an answer to all the points raised by the Reviewer.
